# Prediction model and risk scores of ICU admission and mortality in COVID-19

Zirun Zhao[1☉], Anne Chen[1☉], Wei Hou[1], James M. Graham[1], Haifang Li[1], Paul S. Richman[2], Henry C. Thode[3], Adam J. Singer[3], Tim Q. Duong[1] *

**1** Department of Radiology, Renaissance School of Medicine at Stony Brook University, Stony Brook, New York, United States of America, **2** Division of Pulmonary, Critical Care, and Sleep Medicine, Department of Medicine, Renaissance School of Medicine at Stony Brook University, Stony Brook, New York, United States of America, **3** Department of Emergency Medicine, Renaissance School of Medicine at Stony Brook University, Stony Brook, New York, United States of America

☉ These authors contributed equally to this work.
\* tim.duong@stonybrookmedicine.edu

**Data Availability Statement:** The dataset used in this manuscript is part of a larger database governed by the COVID-19 Research Group Task Force of our institution. We currently do not have

## Abstract

This study aimed to develop risk scores based on clinical characteristics at presentation to predict intensive care unit (ICU) admission and mortality in COVID-19 patients. 641 hospitalized patients with laboratory-confirmed COVID-19 were selected from 4997 persons under investigation. We performed a retrospective review of medical records of demographics, comorbidities and laboratory tests at the initial presentation. Primary outcomes were ICU admission and death. Logistic regression was used to identify independent clinical variables predicting the two outcomes. The model was validated by splitting the data into 70% for training and 30% for testing. Performance accuracy was evaluated using area under the curve (AUC) of the receiver operating characteristic analysis (ROC). Five significant variables predicting ICU admission were lactate dehydrogenase, procalcitonin, pulse oxygen saturation, smoking history, and lymphocyte count. Seven significant variables predicting mortality were heart failure, procalcitonin, lactate dehydrogenase, chronic obstructive pulmonary disease, pulse oxygen saturation, heart rate, and age. The mortality group uniquely contained cardiopulmonary variables. The risk score model yielded good accuracy with an AUC of 0.74 ([95% CI, 0.63–0.85], p = 0.001) for predicting ICU admission and 0.83 ([95% CI, 0.73–0.92], p<0.001) for predicting mortality for the testing dataset. This study identified key independent clinical variables that predicted ICU admission and mortality associated with COVID-19. This risk score system may prove useful for frontline physicians in clinical decision-making under time-sensitive and resource-constrained environment.

## Introduction

The coronavirus disease 2019 (COVID-19) is an infectious disease that can cause severe respiratory illness [1, 2]. First reported in Wuhan, China in December 2019 [3], COVID-19 was declared a pandemic on March 11, 2020 [4]. Currently, more than 3 million people have been infected and more than 300,000 have died of COVID-19 [5]. The actual numbers are believed

approval from the Task Force nor IRB to share any COVID-19 dataset at this time as it is protected patient information. We will work with our institution to enable sharing of these data as soon as possible. Please contact the corresponding author or irb@stonybrook.edu for dataset requests at this time.

**Funding:** The authors received no specific funding for this work.

**Competing interests:** The authors have declared that no competing interests exist.

to be even higher due to testing shortages [6]. Numbers of infection and mortality are predicted to continue to rise in the near future and there are likely to be second waves and future recurrence [7].

There is an urgent need to help frontline clinicians to effectively triage patients in the COVID-19 pandemic. Many patients deteriorate rapidly after a period of relatively mild symptoms, emphasizing the need for early risk stratification [1, 2]. Current literature has already identified several clinical features associated with the severity of COVID-19 infection, but a simple scoring system specific to COVID-19 is lacking [1, 2, 8]. Unfortunately, established early risk scores, such as Sequential Organ Failure Assessment (SOFA) and Modified Early Warning Score (MEWS), have mixed accuracy in predicting COVID-19 severity [9–11]. Here we report a predictive model and a risk score system to predict intensive care unit (ICU) admission and in-hospital mortality in laboratory-confirmed COVID-19 patients. This model was developed and internally validated using data from the COVID-19 persons under investigation (PUI) registry of 4997 patients from a major academic hospital in New York.

## Methods

Stony Brook University Hospital, located about 40 miles east of New York City on Long Island, is the only academic hospital in Suffolk County with a population of approximately 1.5 million. The incidence of COVID-19 infections in Suffolk County has been among the highest in the country [12]. As the pandemic evolved, our hospital reconfigured to increase ICU beds and inpatient capacity. The Army Corps of Engineers had also built five mobile field hospitals on our campus, totaling 1000 additional beds.

### Study population and data collection

This was a retrospective study from Stony Brook University Hospital (March 9, 2020 to April 20, 2020). The study was approved by the Human Subjects Committee with an exemption for informed consent and HIPAA waiver. The COVID-19 PUI registry consisted of 4997 patients. Only hospitalized patients who were diagnosed by positive tests of real-time polymerase chain reaction (RT-PCR) for severe acute respiratory syndrome coronavirus 2 (SARS-CoV-2) were included in this study. Of those, we excluded patients who were still hospitalized because their outcomes were unknown at the time, thus risking grouping them incorrectly. Patients who were younger than 18 years of age, and those with incomplete past medical history were also excluded. One patient who expired very quickly after admission and did not receive any laboratory testing was also excluded. Clinical data at hospital admission, including demographic information, chronic comorbidities, vital signs, symptoms, laboratory tests, and outcomes were collected from the Electronic Medical Record and REDCAP database of our COVID-19 PUI registry [13].

### Outcomes

The primary outcomes were admission to the ICU and death. Among the COVID-19 patients admitted to the hospital, those who were admitted to the ICU met any one of the following criteria: (i) patients developing respiratory failure requiring mechanical ventilation; (ii) patients who have another organ failure requiring ICU monitoring. Our criteria of ICU admission meet the definition of critical illness as described in previous literature of COVID-19, thus we believe it is an appropriate primary outcome for this study [1, 14]. Our mortality group included patients who died during their hospital stay. Both outcomes were compared to a group of patients requiring only general admission, which included patients who were hospitalized and discharged without receiving ICU care at any point of their hospitalization.

## Statistical analysis and modeling

We followed the TRIPOD guideline for developing a multivariable regression model [15]. Categorical variables were described as frequencies and percentages and were compared using $\chi^2$ tests or Fisher exact tests. Continuous variables were presented as medians and interquartile ranges (IQR) and compared using Mann-Whitney U tests. Total data were split into 70% training set and 30% testing set. Data collected from March 9 to April 14, 2020 were used to build the prediction model (n = 454, 70%). Data collected from April 14 to 20, 2020 were used for independent internal validation (n = 187, 30%). We recognized that this choice (instead of random sampling) would likely yield the worst possible performance to challenge our model. Logistic regression models were fit for ICU admission and mortality against general admission as dependent variables, and all available demographic and laboratory variables were included as independent variables. Laboratory variables were transformed to dichotomous variables based on optimal cut off values using the Youden index determined by a nonparametric Kernel regression method for maximizing the summation of sensitivity and specificity [16]. Backward selection was performed and only significant predictors (p<0.05) were kept in the final models. Predictors in the regression model were checked for collinearity and independence. A risk score was developed by assigning each significant variable remaining in the final model to a value of one based on the cut off value. We also developed a weighted risk score model, in which each variable was assigned a weight based on their odds ratio. Prediction performance was evaluated by the area under the curve (AUC) of the receiver operating characteristic (ROC) curve. All analyses were performed in SPSS v26 and SAS v9.4.

## Results

### Patient selection

Fig 1 shows the flowchart of patient selection. The cohort consisted of 4997 PUI over approximately six weeks (March 9 to April 20, 2020), with 1874 confirmed to be COVID-19 positive and 3123 nonconfirmed cases. There were 1232 patients who met the exclusion criteria listed in *Methods*. The final sample size used in this analysis was 641 hospitalized COVID-19 positive patients (median age, 60 years old, 40.1% female), of which 398 did not have critical illness and were admitted to a non-critical care floor, 195 were admitted to the ICU, and 82 who expired. There were 34 patients who died after ICU admission.

### Characteristics of the ICU admission group

As summarized in Table 1, the median age of patients who were admitted to the ICU was 60 years (IQR, 50.0–70.0) and that of those who were not admitted to the ICU (general admission) was 58 years (IQR, 46–71) (p = 0.13). Male gender was significantly associated with higher risk of ICU admission (69.7% vs. 55.8%, p = 0.001). None of the comorbidities were significantly associated with the ICU group (p>0.05). A higher proportion of ICU patients presented with shortness of breath (78.5% vs. 64.8%, p = 0.001), while diarrhea was more common in patients without critical illness (25.6% vs. 17.9%, p = 0.04). Other symptoms on the initial presentation including nausea or vomiting, fever, cough, fatigue, sputum, myalgia, sore throat, rhinorrhea, loss of smell, loss of taste, headache, and chest discomfort were not significantly different between those in the ICU and general hospitalization (p>0.05).

The majority of patients in both the ICU (91.7%) and general admission (81.1%) groups had positive chest x-ray (CXR) findings, but more patients in the ICU group had positive CXR findings (p = 0.001). Similarly, both the ICU group (89.3%) and general admission (78.8%)

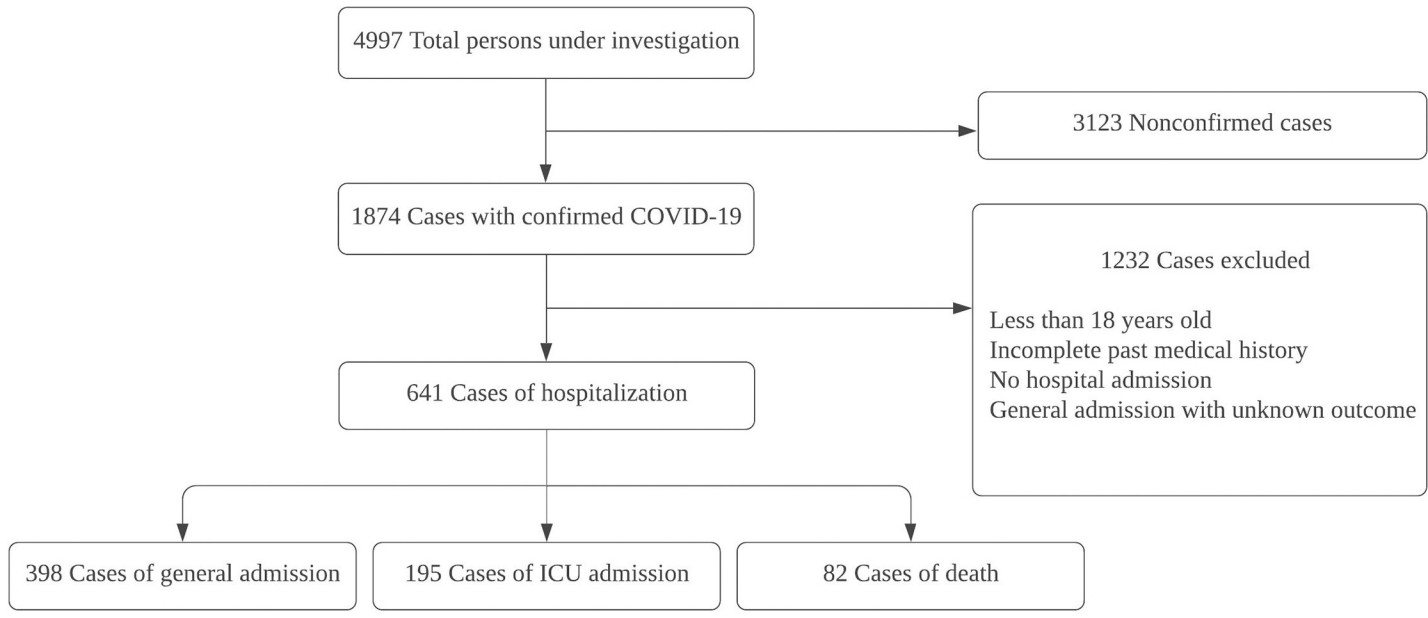

**Fig 1. Flowchart describing patient selection.** Of the 4997 PUIs, 641 hospitalized and laboratory-confirmed COVID-19 patients who did not meet exclusion criteria were included in the study. There was an overlap of 34 patients between ICU and dead group because some patients expired in ICU and others died but did not receive ICU care.

groups also had bilateral involvement on the CXR findings, but more patients in the ICU group had bilateral involvement (p = 0.003).

For vital signs, elevated heart rate (100 vs. 98 rate/min, p = 0.04), elevated respiratory rate (23 vs. 20 rate/min, p<0.001), and decreased pulse oxygen saturation ($SpO_2$) (93% vs. 95%, p<0.001) were significantly associated with ICU admission. However, the magnitude of each of these differences was small and unlikely to be of clinical significance. There was no significant difference in systolic blood pressure and temperature of patients in the ICU compared to those in general admission group (p>0.05).

Compared to the general admission group, the ICU cohort had elevated alanine aminotransferase (ALT) (37 vs. 29 U/L, p = 0.001), brain natriuretic peptide (BNP) (232 vs. 108 ng/L, <0.001), C-reactive protein (CRP) (132 vs. 60 mg/L, p<0.001), D-dimer (2.138 vs. 1.663 nmol/L, p<0.001), ferritin (1149 vs. 582 μg/L, p<0.001), lactate dehydrogenase (LDH) (451 vs. 316 U/L, p<0.001), leukocytes (7.520 vs. 6.770 ×10$^9$/L, p<0.001), and procalcitonin (0.29 vs. 0.08 ng/mL, p<0.001). Patients admitted to the ICU had a decreased fraction of lymphocytes in the peripheral blood (10.9% vs. 14.9%, p<0.001) compared to those in general admission. Cardiac troponin was found to be not significantly different between groups (p>0.05).

## Characteristics of mortality group

The demographics, comorbidities, symptoms, imaging findings, vital signs, and laboratory findings of the mortality group compared with the survival group are shown in Table 2. Of the total 82 patients who died, the median age was 77 years (IQR, 66–85), and 53 (64.6%) were male.

The frequency of several comorbidities was higher in patients who died, compared to those who survived, as follows: smoking history (43.9% vs. 21.9%, p<0.001), hypertension (63.4% vs. 42.7%, p = 0.001), chronic obstructive pulmonary disease (COPD) (18.3% vs. 6.3%, p<0.001), coronary artery disease (30.5% vs. 11.6%, p<0.001), heart failure (26.8% vs. 2.5%, p<0.001), and chronic kidney disease (17.1% vs. 7.1%, p = 0.003).

**Table 1. Demographics, comorbidities, symptoms, imaging findings, vital signs, and laboratory findings of the ICU admission group compared with the general admission group.**

| | Patients, No. (%) | | |
| --- | --- | --- | --- |
| | ICU admission (n = 195) | General admission (n = 398) | p value |
| **Demographics** | | | |
| Age, median (IQR), y | 60 (50, 70) | 58 (46, 71) | 0.13 |
| Sex | | | 0.001 |
| Male | 136 (69.7%) | 222 (55.8%) | |
| Female | 59 (30.3%) | 176 (44.2%) | |
| Ethnicity | | | 0.22 |
| Hispanic/Latino | 56 (28.7%) | 118 (29.6%) | |
| Non-Hispanic/Latino | 98 (50.3%) | 219 (55.0%) | |
| Unknown | 41 (21%) | 61 (15.3%) | |
| Race | | | 0.08 |
| Caucasian | 81 (41.5%) | 190 (47.7%) | |
| African American | 10 (5.1%) | 32 (8%) | |
| Others | 104 (53.3%) | 176 (44.2%) | |
| **Comorbidities** | | | |
| Smoking | 52 (26.7%) | 87 (21.9%) | 0.19 |
| Diabetes | 58 (29.7%) | 104 (26.1%) | 0.35 |
| Hypertension | 96 (49.2%) | 170 (42.7%) | 0.13 |
| Asthma | 16 (8.2%) | 25 (6.3%) | 0.39 |
| COPD | 11 (5.6%) | 25 (6.3%) | 0.76 |
| Coronary artery disease | 22 (11.3%) | 46 (11.6%) | 0.92 |
| Heart failure | 10 (5.1%) | 10 (2.5%) | 0.09 |
| Cancer | 9 (4.6%) | 25 (6.3%) | 0.41 |
| Immunosuppression | 12 (6.2%) | 22 (5.5%) | 0.76 |
| Chronic kidney disease | 16 (8.2%) | 28 (7.1%) | 0.62 |
| **Signs and Symptoms** | | | |
| Fever | 143 (73.3%) | 281 (70.6%) | 0.49 |
| Cough | 148 (75.9%) | 297 (74.6%) | 0.74 |
| Shortness of breath | 153 (78.5%) | 258 (64.8%) | 0.001 |
| Fatigue | 45 (23.1%) | 87 (21.9%) | 0.74 |
| Sputum | 19 (9.7%) | 31 (7.8%) | 0.42 |
| Myalgia | 48 (24.6%) | 109 (27.4%) | 0.47 |
| Diarrhea | 35 (17.9%) | 102 (25.6%) | 0.04 |
| Nausea or vomiting | 32 (16.4%) | 90 (22.6%) | 0.08 |
| Sore throat | 15 (7.7%) | 38 (9.5%) | 0.46 |
| Rhinorrhea | 9 (4.6%) | 19 (4.8%) | 0.93 |
| Loss of smell | 8 (4.1%) | 14 (3.5%) | 0.72 |
| Loss of taste | 9 (4.6%) | 17 (4.3%) | 0.85 |
| Headache | 22 (11.3%) | 45 (11.3%) | 0.99 |
| Chest discomfort | 30 (15.4%) | 63 (15.8%) | 0.89 |
| **Imaging studies** | | | |
| Abnormal chest x-ray results | 177 (91.7%) | 308 (81.1%) | 0.001 |
| Chest x-ray findings | | | 0.003 |
| Unilateral | 19 (10.7%) | 65 (21.2%) | |
| Bilateral | 158 (89.3%) | 242 (78.8%) | |
| **Vital signs, median (IQR)** | | | |

*(Continued)*

**Table 1.** (Continued)

| | Patients, No. (%) | | |
|---|---|---|---|
| | ICU admission (n = 195) | General admission (n = 398) | p value |
| Heart Rate, bpm | 100 (87, 115) | 98 (85, 109) | 0.04 |
| Respiratory rate, rate/min | 23 (18, 28) | 20 (18, 22) | < 0.001 |
| SpO$_2$% | 93 (86, 95) | 95 (93, 97) | < 0.001 |
| SBP, mmHg | 122 (108, 140) | 126 (114, 141) | 0.06 |
| Temperature, ˚C | 37.6 (37.0, 38.5) | 37.4 (36.9, 38.2) | 0.12 |
| **Laboratory findings at admission, median (IQR)** | | | |
| Alanine aminotransferase, U/L | 37 (24, 55) | 29 (18, 51) | 0.001 |
| Brain natriuretic peptide, ng/L | 232 (74, 883) | 108 (32, 483) | < 0.001 |
| C-reactive protein, mg/L | 132 (78, 215) | 60 (27, 120) | < 0.001 |
| D-dimer, nmol/L | 2.138 (1.345, 3.645) | 1.663 (1.048, 2.689) | < 0.001 |
| Ferritin, µg/L | 1149 (605, 1901) | 582 (261, 1140) | < 0.001 |
| Lactate dehydrogenase, U/L | 451 (342, 609) | 316 (247, 398) | < 0.001 |
| Leukocytes×10$^9$/liter | 7.520 (5.905, 10.245) | 6.770 (5.110, 8.720) | < 0.001 |
| Lymphocytes % | 10.9 (6.8, 16.0) | 14.9 (10.0, 21.7) | <0.001 |
| Procalcitonin, ng/mL | 0.29 (0.16, 0.73) | 0.08 (0.06, 0.25) | < 0.001 |
| Troponin, µg/L | 0.01 (0.01, 0.01) | 0.01 (0.01, 0.01) | 0.19 |

Abbreviation: ICU, intensive care unit. COPD, chronic obstructive pulmonary disease. IQR, interquartile range. SpO$_2$, pulse oxygen saturation.

**Table 2. Demographics, comorbidities, symptoms, imaging findings, vital signs, and laboratory findings of the mortality group compared with the survival group.**

| | Patients, No. (%) | | |
|---|---|---|---|
| | Died (n = 82) | Survived (n = 398) | p value |
| **Demographics** | | | |
| Age, median (range), y | 77 (66, 85) | 58 (46, 71) | < 0.001 |
| Sex | | | 0.14 |
| Male | 53 (64.6%) | 222 (55.8%) | |
| Female | 29 (35.4%) | 176 (44.2%) | |
| Ethnicity | | | < 0.001 |
| Hispanic/Latino | 8 (9.8%) | 118 (29.6%) | |
| Non-Hispanic/Latino | 63 (76.8%) | 219 (55%) | |
| Unknown | 11 (13.4%) | 61 (15.3%) | |
| Race | | | 0.01 |
| Caucasian | 53 (64.6%) | 190 (47.7%) | |
| African American | 7 (8.5%) | 32 (8%) | |
| Others | 22 (26.8%) | 176 (44.2%) | |
| **Comorbidities** | | | |
| Smoking history | 36 (43.9%) | 87 (21.9%) | < 0.001 |
| Diabetes | 25 (30.5%) | 104 (26.1%) | 0.42 |
| Hypertension | 52 (63.4%) | 170 (42.7%) | 0.001 |
| Asthma | 3 (3.7%) | 25 (6.3%) | 0.45 |
| COPD | 15 (18.3%) | 25 (6.3%) | < 0.001 |
| Coronary artery disease | 25 (30.5%) | 46 (11.6%) | < 0.001 |
| Heart failure | 22 (26.8%) | 10 (2.5%) | < 0.001 |
| Cancer | 7 (8.5%) | 25 (6.3%) | 0.5 |

*(Continued)*

**Table 2.** (Continued)

| | Patients, No. (%) | | |
|---|---|---|---|
| | **Died (n = 82)** | **Survived (n = 398)** | **p value** |
| Immunosuppression | 8 (9.8%) | 22 (5.5%) | 0.15 |
| Chronic kidney disease | 14 (17.1%) | 28 (7.1%) | 0.003 |
| **Symptoms** | | | |
| Fever | 48 (58.5%) | 281 (70.6%) | 0.03 |
| Cough | 46 (56.1%) | 297 (74.6%) | 0.001 |
| Shortness of breath | 57 (69.5%) | 258 (64.8%) | 0.42 |
| Fatigue | 12 (14.6%) | 87 (21.9%) | 0.14 |
| Sputum | 8 (9.8%) | 31 (7.8%) | 0.56 |
| Myalgia | 5 (6.1%) | 109 (27.4%) | < 0.001 |
| Diarrhea | 10 (12.2%) | 102 (25.6%) | 0.009 |
| Nausea or vomiting | 2 (2.4%) | 90 (22.6%) | < 0.001 |
| Sore throat | 3 (3.7%) | 38 (9.5%) | 0.08 |
| Rhinorrhea | 2 (2.4%) | 19 (4.8%) | 0.55 |
| Loss of smell | 0 (0%) | 14 (3.5%) | 0.14 |
| Loss of taste | 0 (0%) | 17 (4.3%) | 0.09 |
| Headache | 4 (4.9%) | 45 (11.3%) | 0.08 |
| Chest discomfort or chest pain | 5 (6.1%) | 63 (15.8%) | 0.02 |
| **Imaging studies** | | | |
| Abnormal chest x-ray results | 65 (83.3%) | 308 (81.1%) | 0.64 |
| Chest x-ray findings | | | 0.18 |
| Unilateral | 9 (13.8%) | 65 (21.2%) | |
| Bilateral | 56 (86.2%) | 242 (78.8%) | |
| **Vital signs, median (IQR)** | | | |
| Heart Rate, bpm | 97 (84, 115) | 98 (85, 109) | 0.81 |
| Respiratory rate, rate/min | 24 (18, 29) | 20 (17, 22) | < 0.001 |
| SpO$_2$% | 93 (87, 96) | 95 (93, 97) | < 0.001 |
| Systolic blood pressure, mmHg | 129 (107, 145) | 126 (114, 141) | 0.88 |
| Temperature, ˚C | 37.1 (36.7, 37.7) | 37.4 (36.9, 38.2) | 0.004 |
| **Laboratory findings at admission, median (IQR)** | | | |
| Alanine aminotransferase, U/L | 29 (17, 50) | 29 (18, 51) | 0.97 |
| Brain natriuretic peptide, ng/L | 1583 (397, 4229) | 108 (32, 483) | < 0.001 |
| C-reactive protein, mg/L | 139 (70, 211) | 60 (27, 120) | < 0.001 |
| D-dimer, nmol/L | 3.537 (1.966, 9.963) | 1.663 (1.048, 2.689) | < 0.001 |
| Ferritin, µg/L | 843 (417, 1526) | 582 (261, 1140) | 0.005 |
| Lactate dehydrogenase, U/L | 440 (293, 618) | 316 (247, 398) | < 0.001 |
| Leukocytes ×10$^9$/liter | 8.260 (5.970, 10.490) | 6.770 (5.110, 8.720) | 0.002 |
| Lymphocytes% | 8.8 (5.4, 14.8) | 14.9 (10.0, 21.7) | < 0.001 |
| Procalcitonin, ng/mL | 0.33 (0.16, 1.34) | 0.13 (0.08, 0.25) | < 0.001 |
| Troponin, µg/L | 0.02 (0.01, 0.06) | 0.01 (0.01, 0.01) | < 0.001 |

Abbreviation: COPD, chronic obstructive pulmonary disease. IQR, interquartile range. SpO$_2$, pulse oxygen saturation.

Interestingly, the patients who survived had more self-reported symptoms compared to those who died, as follows: fever (70.6% vs. 58.5%, p = 0.03) cough (74.6% vs. 56.1%, p = 0.001) myalgia (27.4% vs. 6.1%, p<0.001), diarrhea (25.6% vs. 12.2%, p = 0.009), nausea or vomiting (22.6% vs. 2.4%, p<0.001), and chest discomfort (15.8% vs. 6.1%, p = 0.02).

The majority of patients had abnormalities on CXR in both the deceased (83.3%) and survivors (81.1%), but there was no statistical difference between groups (p>0.05). Similarly, the majority of patients had bilateral involvement in both the mortality group (86.2%) and survival group (78.8%), also with no statistical difference between groups (p>0.05).

For vital signs, increased respiratory rate (24 vs. 20 rate/min, p<0.001), decreased SpO$_2$ (93% vs. 95%, p<0.001), and decreased temperature (37.1 vs. 37.4˚C, p = 0.004) were associated with mortality. Again, the magnitude of these mean differences in vital signs was small and unlikely to be clinically meaningful.

Compared to the patients who survived, those who died had elevated BNP (1583 vs. 108 ng/L, p<0.001), CRP (139 vs. 60 mg/L, p<0.001), D-dimer (3.537 vs. 1.663 nmol/L, p<0.001), ferritin (843 vs. 582 µg/L, p = 0.005), LDH (440 vs. 316 U/L, p<0.001), leukocytes (8.260 vs. 6.770 ×10$^9$/liter, p<0.002), procalcitonin (0.33 vs. 0.13 ng/mL, p<0.001), and cardiac troponin (0.02 vs. 0.01 µg/L, p<0.001). Percent lymphocytes in the peripheral blood were lower in the patients who died (8.8% vs. 14.9%, p<0.001).

## Top predictors of ICU admission and mortality

Our logistic regression model developed using the training dataset (n = 454) identified the top five ICU-admission predictors to be LDH (Odds Ratio [OR] 3.3, [95% CI, 1.89–5.88]), procalcitonin (OR 2.77, [95% CI, 1.57–4.89]), smoking history (OR 2.23, [95% CI, 1.17–4.27]), SpO$_2$ (OR 1.90, [95% CI, 1.07–3.37]), and lymphocyte count (OR 1.83, [95% CI, 1.04–3.22]). The top seven mortality predictors were heart failure (OR 33.48, [95% CI, 4.99–224.45]), procalcitonin (OR 6.31, [95% CI, 0.79–22.26]), LDH (OR 5.78, [95% CI, 1.65–20.28]), COPD (OR 9.23, [95% CI, 1.89–45.01]), SpO$_2$ (OR 4.80, [95% CI, 1.32–17.45]), heart rate (OR 7.73, [95% CI, 1.27–46.90]), and age (OR 4.90, [95% CI, 1.17–20.50]) (Table 3). Some of the top predictors were common to both the ICU admission group and the mortality group, but the mortality group uniquely contained cardiopulmonary parameters (i.e., history of heart failure, COPD, elevated heart rate) amongst its top predictors.

## Risk scores for ICU admission and mortality

We then developed a risk score to predict ICU admission and mortality in COVID-19 positive patients from the top predictors we identified (Fig 2). The risk score for predicting ICU

**Table 3. Top variables predicting ICU admission and mortality against general admission and survival, respectively.**

| Variables predicting ICU admission | Odds ratio (95% CI) | p value |
|---|---|---|
| LDH (>389 U/L) | 3.34 (1.89, 5.88) | <0.001 |
| Procalcitonin (>0.22 ng/mL) | 2.77 (1.57, 4.89) | <0.001 |
| Smoking history, ever smoker | 2.23 (1.17, 4.27) | 0.02 |
| SpO$_2$ (<92%) | 1.90 (1.07, 3.37) | 0.03 |
| Lymphocyte count (<12%) | 1.83 (1.04, 3.22) | 0.04 |
| **Variables predicting mortality** | **Odds ratio (95% CI)** | **p value** |
| Heart failure | 33.48 (4.99, 224.45) | <0.001 |
| Procalcitonin (>0.34 ng/mL) | 6.31 (1.79, 22.26) | 0.004 |
| LDH (>460 U/L) | 5.78 (1.65, 20.28) | 0.006 |
| COPD | 9.23 (1.90, 45.01) | 0.006 |
| SpO$_2$ (<92%) | 4.80 (1.32, 17.45) | 0.02 |
| Heart rate (>117 bpm) | 7.73 (1.27, 46.90) | 0.03 |
| Age (>63 years) | 4.90 (1.17, 20.50) | 0.03 |

Abbreviations: LDH, lactate dehydrogenase. SpO$_2$, pulse oxygen saturation. COPD, chronic obstructive pulmonary disease.

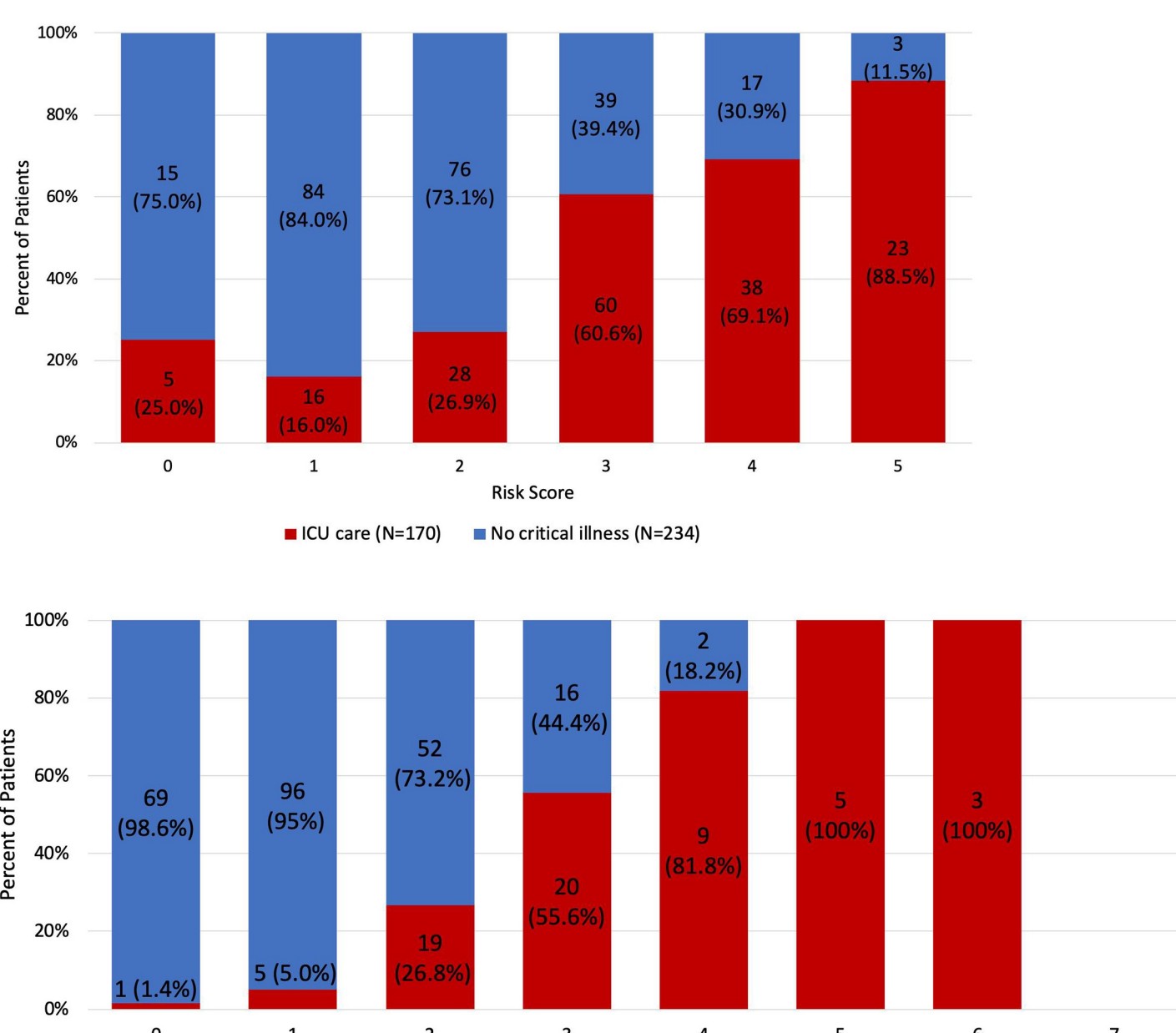

**Fig 2. Risk score stratifications for ICU admission and for mortality.** (A) Risk score stratification for ICU admission. (B) Risk score stratification for mortality. There were no patients with risk score of 7 in the mortality group.

admission ranged from 0 to 5, from lowest to highest risk. The score was calculated by assigning 1 point each for the five top predictors of ICU admission in Table 3. The risk score for predicting mortality ranged from 0 to 7, from lowest to highest risk. The score was calculated by assigning 1 point for each of the seven top predictors of mortality (Table 3). The percentage of patients in ICU care increased with increasing ICU-admission risk score, while the percentage of patients in non-ICU care decreased with increased risk score. Similarly, the percentage of

deceased patients increased with increasing mortality risk score, and the percentage of patients who survived decreased with increased risk score. Prediction performance in the training set yielded an AUC of 0.761 ([95% CI, 0.71–0.81], p<0.001) for ICU admission and 0.87, ([95% CI, 0.83–0.92], p<0.001) for mortality (n = 454). The AUC was 0.74 ([95% CI, 0.63–0.85], p = 0.001) for predicting ICU admission, and 0.82 ([95% CI, 0.73–0.92], p<0.001) for predicting mortality for the test dataset (n = 187). The sensitivity and specificity of the risk scores were 10.5% and 99.2% for predicting ICU admission, and 7.1% and 100% to predict mortality. The high specificity and low sensitivity were due to the imbalance of sample sizes where there were more patients in general admission group compared to ICU admission or death group. We also developed a weighted risk score model, where each variable was assigned a weight based on their odds ratio. However, the AUC was comparable to the non-weighted model. Consequently, we determined that the non-weighted model was superior because of its simplicity and good accuracy.

## Discussion

The present study used data obtained early in the course of COVID-19 infection to develop simple risk scores that elucidate two issues: 1. Providing objective evidence to aid in the decision of ICU admission, and 2. Quantifying the risk of mortality from this infection, based on parameters during their initial (emergency department) presentation. To our knowledge, this is the first risk score developed using a large US sample. The strengths of our study include rigorous methods for developing multivariable regression models and a separate dataset for validation of our risk scores. The simplified risk score system yielded an AUC of 0.74 for predicting ICU admission, and 0.82 for predicting mortality for the testing dataset. These tools can help direct COVID-19 patient flow and appropriately allocate resources.

The common top predictors of ICU admission and mortality were elevated LDH and procalcitonin, and reduced SpO$_2$. Elevated LDH indicates cell death and injury and is associated with a poor host immune response, resulting in a higher susceptibility to severe viral infections [17–19]. Procalcitonin is a widely-used indicator for critical illness due to infectious etiology, albeit with controversy [20]. It has been reported that viral infections induce the release of inflammatory cytokines such as interferon-γ which decreases procalcitonin levels [21]. Substantial increases in procalcitonin during a viral infection may therefore indicate a deficient immune response to clear the infection [21]. Severe respiratory failure and death caused by COVID-19 may result from damaged alveoli and edema formation, negatively affecting the lung's ability to oxygenate blood, as reflected in reduced oxygen saturation [22, 23]. As arterial blood-gas measurement is invasive and not regularly performed at the first point of care, SpO$_2$ serves as a more available indicator for oxygenation for triage purposes.

Some variables were uniquely associated with either ICU admission or mortality. Reduced lymphocyte count was amongst the top predictors of ICU admission but not of mortality. Lymphopenia is a common characteristic of infection caused by the body's cytokine-induced reaction [24]. The reduced CD4+ T-cell and CD8+ T-cell levels promote viral survival and predict worse outcome [25]. In terms of past medical history and comorbidities, smoking history was a unique top predictor of ICU admission but was not associated with increased mortality in our cohort. However, medical histories of heart disease, COPD, and elevated heart rate at presentation were unique top predictors of mortality but not ICU admission, suggesting that these cardiopulmonary risk factors may contribute more to mortality in COVID-19 patients. Our findings were consistent with a recent study that found SARS-CoV-2 to cause myocardial injury through mechanisms of inflammation, resulting in fatal cardiac dysfunction and arrythmias [26].

Interestingly, unlike previous literature which generally cited age as an important predictor of mortality, our model did not find age to be amongst the top predictors for mortality in our cohort [27, 28]. These results suggested that comorbidities associated with aging, rather than advanced age itself, contribute to a worse prognosis. CXR findings in our cohort were not predictive of ICU admission or mortality. However, our imaging analysis included only the presence of any abnormalities and the laterality of lung involvement. Detailed imaging findings, such as ground glass opacity and consolidation, and radiologist scoring of disease severity, is under investigation. It is not surprising that symptoms were not amongst the top predictors. Although some were significantly different between groups, it was likely attributed to the subjective reporting and non-specific nature of symptoms such as fever, cough, and shortness of breath. Ethnicity and race were not amongst our top predictors although further studies are warranted [29, 30].

It is also possible to construct a model to classify hospitalized and non-hospitalized COVID-19 patients. However, data from non-hospitalized patients were limited and consisted mainly of vitals and demographics, usually without laboratory and imaging tests. A recent paper from our hospital supported this finding. In that study, Singer et al. found that confirmed COVID-19 patients had worse outcomes than COVID-19 negative patients [13]. Our study instead focused only on COVID-19 positive patients and covered different time periods, explaining some differences in the predictor variables identified from the prior study at our institution.

A few studies have attempted to develop a predictive or risk score model to facilitate clinical decision making associated with COVID-19 patients. General risk scores, such as SOFA and MEWS, lack sensitivity and specificity to predict mortality when applied to COVID-19 infection [9, 10]. Lu et al. created a three-tiered risk score based on only two variables, age and CRP thresholds, to determine mortality [28]. Ji et al. identified comorbidities, age, lymphocyte count and LDH to be predictors of mortality [27]. Xie et al. reported age, lymphocyte count, LDH and $SpO_2$ to be independent predictors of mortality but a risk score was not developed [31]. None of these previous studies statistically ranked the predictors nor validated the risk scores using independent datasets. While Zhou et al. developed a nomogram for predicting severity of COVID-19 from comorbidities and symptoms, they did not include any laboratory values, which is an important component in the clinical realm [32]. All these prediction models were also only based on Chinese patient cohorts, who have different health profiles to US cohorts. As a result, our model may have more implications in US hospitals, which are experiencing a later surge than the rest of the world. Variables like smoking history, COPD and heart failure for example, were not part of their models, but play a significant role in our risk score.

While our model has some overlapping variables with previous risk scores, the variability between our studies can be attributed to different patient cohorts with different lifestyles, different outcomes measures being investigated (mortality, ICU admission, and disease severity), hospital environments, and statistical methods employed, among other factors. Thus, a good predictive model needs to be flexible to incorporate new and local data. Unlike most of the previous predictive models, our model was validated internally using independent datasets. Our risk-score model also incorporated 5 to 7 significant predictors, more than those in previously studies but still a manageable amount to paint an accurate clinical picture without being too broad.

This study had several limitations. First, while our risk score identified five predictors of ICU admission, it does not mean these should be the only criteria for determining ICU admission. These predictors should instead be used in conjunction to support a clinician's decision to admit a patient to the ICU. We also recognized that our hospital's criteria for ICU

admission can differ from that of other hospitals. In addition to how our study measured mortality, we realized that there are other measures, such as a "time-to-event" approach or fixed follow-up period. This was also a retrospective study based on a single institution. We will collaborate with other institutions to test our model to improve generalizability. This study included clinical data only at presentation. Incorporating longitudinal clinical data are currently under investigation. While this prediction model was based on logistic regression, machine learning and other methods are being explored. Finally, it is important to note that the COVID-19 pandemic circumstance is unusual and evolving. Flow of patients (i.e., to general admission or ICU) and mortality may depend on individual hospital's patient load, practice, and available resources, which also differ amongst countries. Thus, this prediction model may need to be retrained with regional data, which can be readily accomplished.

## Conclusion

This study identified the highly ranked clinical features that accurately predict ICU admission and mortality associated with COVID-19 infection. Based on these findings, we developed a practical risk-score model to stratify patients into general versus ICU admission, and to predict mortality. Our model can be readily enhanced or retrained using additional data and data from other institutions. This approach has the potential to provide frontline physicians with a simple and objective tool to stratify patients based on risks so that they can triage COVID-19 patients more effectively in time-sensitive, stressful and potentially resource-constrained environments.

## Acknowledgments

We thank all healthcare professionals for their hard work being at the front line of the pandemic. All authors declare no conflict of interest, including financial interests, activities, relationships, and affiliations. This research did not receive any specific grant from funding agencies in the public, commercial, or not-for-profit sectors.

## Author Contributions

**Conceptualization:** Zirun Zhao, Anne Chen, Tim Q. Duong.

**Data curation:** Zirun Zhao, Anne Chen, James M. Graham, Haifang Li, Henry C. Thode, Adam J. Singer, Tim Q. Duong.

**Formal analysis:** Zirun Zhao, Anne Chen, Wei Hou.

**Investigation:** Zirun Zhao, Haifang Li, Tim Q. Duong.

**Methodology:** Zirun Zhao, Anne Chen, Wei Hou, Tim Q. Duong.

**Project administration:** Haifang Li, Henry C. Thode, Adam J. Singer, Tim Q. Duong.

**Resources:** Haifang Li, Henry C. Thode, Tim Q. Duong.

**Software:** Haifang Li.

**Supervision:** Wei Hou, Haifang Li, Paul S. Richman, Adam J. Singer, Tim Q. Duong.

**Validation:** Zirun Zhao, Anne Chen, Wei Hou, Haifang Li, Paul S. Richman, Henry C. Thode, Adam J. Singer, Tim Q. Duong.

**Visualization:** Zirun Zhao, Anne Chen, Wei Hou.

**Writing – original draft:** Zirun Zhao, Anne Chen, Tim Q. Duong.

**Writing – review & editing:** Zirun Zhao, Anne Chen, Wei Hou, James M. Graham, Paul S. Richman, Adam J. Singer, Tim Q. Duong.

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
