## [Decision Letter · Decision Letter 0]

3 Jul 2020

PONE-D-20-15746

Prediction model and risk scores of ICU admission and mortality in COVID-19

PLOS ONE

Dear Dr. Duong,

Thank you for submitting your manuscript to PLOS ONE. After careful consideration, we feel that it has merit but does not fully meet PLOS ONE’s publication criteria as it currently stands. Therefore, we invite you to submit a revised version of the manuscript that addresses the points raised during the review process.

 Please review the comments by the reviewers and provide point by point response in your revised manuscript.

We look forward to receiving your revised manuscript.

Kind regards,

Muhammad Adrish

Academic Editor

PLOS ONE

Journal Requirements:

Reviewers' comments:

Reviewer's Responses to Questions

**Comments to the Author**

1. Is the manuscript technically sound, and do the data support the conclusions?

Reviewer #1: Yes

Reviewer #2: Yes

2. Has the statistical analysis been performed appropriately and rigorously? 

Reviewer #1: Yes

Reviewer #2: Yes

3. Have the authors made all data underlying the findings in their manuscript fully available?

Reviewer #1: No

Reviewer #2: Yes

4. Is the manuscript presented in an intelligible fashion and written in standard English?

Reviewer #1: Yes

Reviewer #2: Yes

5. Review Comments to the Author

Reviewer #1: This study aimed at developing risk scores based on clinical characteristics at presentation to predict ICU admission and mortality in COVID-19 patients.

The paper is well written and show interesting data. However, the authors report just the AUCs of scores. Sensitivity, specificity and likelihood ratios of the proposed scores must be given.

Reviewer #2: In this article by Dr. Timothy Duong et al they aim to develop risk scores based on clinical characteristics at

presentation to predict intensive care unit (ICU) admission and mortality in COVID-19 patients.

According to the authors, five significant variables predict ICU admission (lactate dehydrogenase,

procalcitonin, pulse oxygen saturation, smoking history, and lymphocyte count) and seven

variables predict mortality (heart failure, procalcitonin, lactate dehydrogenase,

chronic obstructive pulmonary disease pulse oxygen saturation, heart rate, and age).

The model predicts ICU admission and a mortality with an AUC of 0.74 and 0.83, respectively.

This article is well written and addresses a very important issue regarding patient flow organization and allocation of resources in the COVID-19 era.

My main comments on the manuscripts have been addressed by the authors in the limitation section. Im looking forward to a follow-up study were authors add more institutions and make the study more generalizable.

I believe this article deserves to be published.

LINE 40: There is a comma missing between COPD and Pulse oxygen

6. PLOS authors have the option to publish the peer review history of their article (what does this mean?). If published, this will include your full peer review and any attached files.

Reviewer #1: No

Reviewer #2: No

---

## [Author Response · Author response to Decision Letter 0]

7 Jul 2020

Thank you for your careful review of our manuscript

Reviewers' comments:

Reviewer's Responses to Questions

Comments to the Author

1. Is the manuscript technically sound, and do the data support the conclusions?

Reviewer #1: Yes

Reviewer #2: Yes

2. Has the statistical analysis been performed appropriately and rigorously? 

Reviewer #1: Yes

Reviewer #2: Yes

3. Have the authors made all data underlying the findings in their manuscript fully available?

Reviewer #1: No

Reviewer #2: Yes

4. Is the manuscript presented in an intelligible fashion and written in standard English?

Reviewer #1: Yes

Reviewer #2: Yes

5. Review Comments to the Author

Reviewer #1: This study aimed at developing risk scores based on clinical characteristics at presentation to predict ICU admission and mortality in COVID-19 patients. The paper is well written and show interesting data. However, the authors report just the AUCs of scores. Sensitivity, specificity and likelihood ratios of the proposed scores must be given.

Thank you. 

The sensitivity and specificity of the risk scores were 10.5% and 99.2% for predicting ICU admission, and 7.1% and 100% to predict mortality. The high specificity and low sensitivity were due to the imbalance of sample sizes where there were more patients in general admission group compared to ICU admission or death group. 

Reviewer #2: In this article by Dr. Timothy Duong et al they aim to develop risk scores based on clinical characteristics at presentation to predict intensive care unit (ICU) admission and mortality in COVID-19 patients. According to the authors, five significant variables predict ICU admission (lactate dehydrogenase, procalcitonin, pulse oxygen saturation, smoking history, and lymphocyte count) and seven variables predict mortality (heart failure, procalcitonin, lactate dehydrogenase, chronic obstructive pulmonary disease pulse oxygen saturation, heart rate, and age). The model predicts ICU admission and a mortality with an AUC of 0.74 and 0.83, respectively. This article is well written and addresses a very important issue regarding patient flow organization and allocation of resources in the COVID-19 era.

My main comments on the manuscripts have been addressed by the authors in the limitation section. Im looking forward to a follow-up study were authors add more institutions and make the study more generalizable.

I believe this article deserves to be published.

Thank you.

LINE 40: There is a comma missing between COPD and Pulse oxygen

Thank you

---

## [Editor Report · Decision Letter 1]

13 Jul 2020

Prediction model and risk scores of ICU admission and mortality in COVID-19

PONE-D-20-15746R1

Dear Dr. Duong,

We’re pleased to inform you that your manuscript has been judged scientifically suitable for publication and will be formally accepted for publication once it meets all outstanding technical requirements.

Kind regards,

Muhammad Adrish

Academic Editor

PLOS ONE

Additional Editor Comments (optional):

Thank you for revising your manuscript. You have satisfactorily answered all reviewer queries.
---

## [Editor Report · Acceptance letter]

21 Jul 2020

PONE-D-20-15746R1 

Prediction model and risk scores of ICU admission and mortality in COVID-19 

Dear Dr. Duong:

I'm pleased to inform you that your manuscript has been deemed suitable for publication in PLOS ONE. Congratulations! Your manuscript is now with our production department. 

Kind regards, 

on behalf of

Dr. Muhammad Adrish 

Academic Editor

PLOS ONE